# Development of a New Ankle Joint Hybrid Assistive Limb

**DOI:** 10.3390/medicina58030395

**Published:** 2022-03-07

**Authors:** Shigeki Kubota, Hideki Kadone, Yukiyo Shimizu, Masao Koda, Hiroshi Noguchi, Hiroshi Takahashi, Hiroki Watanabe, Yasushi Hada, Yoshiyuki Sankai, Masashi Yamazaki

**Affiliations:** 1Department of Orthopaedic Surgery, Faculty of Medicine, University of Tsukuba, Tsukuba 305-8575, Ibaraki, Japan; masaokod@gmail.com (M.K.); noguhiro0164@tsukuba-seikei.jp (H.N.); hhtaka@md.tsukuba.ac.jp (H.T.); masashiy@md.tsukuba.ac.jp (M.Y.); 2Center for Innovating Medicine and Engineering (CIME), University of Tsukuba Hospital, Tsukuba 305-8573, Ibaraki, Japan; kadone@md.tsukuba.ac.jp; 3Department of Rehabilitation Medicine, University of Tsukuba, Tsukuba 305-8575, Ibaraki, Japan; shimiyukig@md.tsukuba.ac.jp (Y.S.); y-hada@md.tsukuba.ac.jp (Y.H.); 4Department of Neurosurgery, Faculty of Medicine, University of Tsukuba, Tsukuba 305-8575, Ibaraki, Japan; watanabe.hiroki.gb@u.tsukuba.ac.jp; 5Faculty of Systems and Information Engineering, University of Tsukuba, Tsukuba 305-8575, Ibaraki, Japan; sankai@golem.iit.tsukuba.ac.jp

**Keywords:** robotic ankle rehabilitation, ankle joint hybrid assistive limb, foot drop

## Abstract

Foot and ankle disabilities (foot drop) due to common peroneal nerve palsy and stroke negatively affect patients’ ambulation and activities of daily living. We developed a novel robotics ankle hybrid assistive limb (HAL) for patients with foot drop due to common peroneal nerve palsy or stroke. The ankle HAL is a wearable exoskeleton-type robot that is used to train plantar and dorsiflexion and for voluntary assistive training of the ankle joint of patients with palsy using an actuator, which is placed on the lateral side of the ankle joint and detects bioelectrical signals from the tibialis anterior (TA) and gastrocnemius muscles. Voluntary ankle dorsiflexion training using the new ankle HAL was implemented in a patient with foot drop due to peroneal nerve palsy after lumbar surgery. The time required for ankle HAL training (from wearing to the end of training) was approximately 30 min per session. The muscle activities of the TA on the right were lower than those on the left before and after ankle HAL training. The electromyographic wave of muscle activities of the TA on the right was slightly clearer than that before ankle HAL training in the resting position immediately after ankle dorsiflexion. Voluntary ankle dorsiflexion training using the novel robotics ankle HAL was safe and had no adverse effect in a patient with foot drop due to peroneal nerve palsy.

## 1. Introduction

Common peroneal nerve palsy and ankle-foot disabilities due to stroke (cerebrovascular accident) can cause foot drop, which disturbs gait and negatively affects activities of daily living [1,2]. Peroneal neuropathy at the neck of the fibula, anterior horn cell disease, lumbar plexopathies, L5 radiculopathy, and partial sciatic neuropathy can cause common peroneal nerve palsy, which leads to foot drop [3]. Conservative treatment strategies using orthotic and physical therapy are often the treatment of choice. However, surgical interventions are recommended if conservative treatment is ineffective, and available methods include neurolysis or tendon transfer [4]. Nonetheless, some patients permanently suffer from foot drop. In contrast, stroke causes hemiplegia, a serious and disabling health problem that is common worldwide [5]. Most patients with stroke experience ankle-foot disability (spasticity) [6]. Therefore, we aimed to improve ankle-foot palsy and the power of dorsiflexors of the foot in patients with foot drop due to common peroneal nerve palsy and to decrease ankle spasticity due to ankle disturbance caused by stroke.

The hybrid assistive limb (HAL) is a wearable exoskeleton robot that provides real-time assistance to a wearer’s motion, such as walking and limb movements via actuators mounted on both the hip and knee joints and bioelectrical signals (muscle action potentials) that detect the electrical activities in the lower limb muscles via surface electrode sensors [7]. There are several types of HAL robots, such as bilateral-limb [8], lateral-limb [9], single-joint [10,11,12], and lumbar robots [13]. We developed an ankle joint HAL for patients with ankle-foot impairment. To the best of our knowledge, there has been no report on the development of ankle joint HALs. Therefore, we aimed to introduce a new ankle HAL and assess its use in the rehabilitation of plantar and dorsiflexion. 

## 2. Materials and Methods

### 2.1. Development and Structure of the Ankle HAL

The ankle HAL system consists of a control device, HAL shoe, leg support, lateral ankle plate, actuator, surface electrode sensor, manual controller, and battery (Figure 1). First, we developed a metal leg support and a lateral ankle plate. Afterward, the leg support was attached proximally to the conventional single-joint HAL. The lateral ankle plate was then attached distally to the conventional single-joint HAL and HAL shoes (Figure 1). The ankle HAL used in this study is a robotic system that is used to support the ankle joint, and we added our originally produced parts (leg support and ankle lateral plate) to a conventional single-joint HAL (shoulder, elbow, and knee).

### 2.2. Procedure and Function of the Ankle HAL

The ankle HAL is a wearable exoskeleton-type robot that is used to train plantar and dorsiflexion and for voluntary assistive training of the ankle joint of patients with palsy using an actuator, which is placed on the lateral side of the ankle joint and detects bioelectrical signals (muscle action potentials) from the tibialis anterior (TA) and gastrocnemius muscles. The actuator consists of an angular sensor on the lateral ankle joint. These various biologic signals and information detected by the angular sensor are processed by a control device (computer). The HAL provides the wearer with a feedback system, allowing them to view their own bioelectrical signals on a monitor in real time during ankle HAL training. First, the wearer was placed in the sitting position on the bed. While the wearer remained in this position, the surface electrode sensors of the HAL were then attached to the TA and gastrocnemius muscle. Next, the surface electrode sensors were connected to the control device, and the therapists turned on the ankle HAL. Subsequently, the ankle HAL with leg support, lateral ankle plate, and HAL shoes were attached to the wearer’s ankle while they remained in the sitting position. The anterior and posterior cushions were placed between the leg support and the wearer’s leg to avoid skin abrasion and were affixed using rubber bands. After completely attaching the ankle HAL, the therapists regulated the movement and level of assistance provided by the HAL using a manual controller during the ankle HAL exercises (Appendix A). Attachment of the ankle HAL requires assistance from two persons and takes approximately 3 min to complete.

The main function of the ankle HAL is the generation of voluntary ankle movements in the wearer using bioelectrical signals (muscle action potentials). Even if the wearer has muscle weakness of the TA (e.g., manual muscle testing; MMT 1) or ankle-joint palsy due to stroke, the ankle HAL can still generate voluntary ankle movements (ankle plantar and dorsiflexion) and detects only bioelectrical signals from the TA or gastrocnemius muscle. Therefore, the ankle HAL can be used to perform ankle exercises to prevent ankle joint stiffness in the acute phase of common peroneal nerve palsy, even if the MMT of the TA is 1 (foot drop).

### 2.3. Case Presentation

We present the case of a 55-year-old woman with a body mass index of 36.9 kg/m^2^. She had low back pain, which started 1 year and 3 months preoperatively. Her condition was diagnosed as ossification of the posterior longitudinal ligament (OPLL) of the lumbar spine and stenosis. The patient had gait disturbance that progressed gradually; therefore, she underwent lateral lumbar interbody fusion at L3/4/5, posterior instrumented fusion at L3–5, and transforaminal lumbar interbody fusion at L5/S1 for lumbar OPLL and lumbar spinal canal stenosis, in two stages. The day after the operation, the patient had weakness in the right TA (MMT, 0–1), extensor hallucis longus (EHL; MMT, 0–1), extensor digitorum longus (MMT, 2), and ultimately had right foot drop (Figure 2a). No hematoma or screw failure was observed on computed tomography images. Assessment of nerve conduction velocity and needle electromyography was performed on postoperative day 26, and a right postoperative common peroneal nerve palsy with foot drop was diagnosed (Figure 2b). The preoperative MMT of the right TA and EHL was 5 each. Although conventional physical therapy rehabilitation started after surgery, the right foot drop was not satisfactorily ameliorated, and walking training was difficult because of other factors, such as obesity and postoperative delirium. Therefore, ankle HAL training was initiated on postoperative day 33.

### 2.4. Dorsiflexion Training Using the Ankle HAL

Right ankle dorsiflexion training using the ankle HAL started on postoperative day 33 (Figure 2c,d and Appendix A). First, the patient was placed in a sitting position and the therapist attached the ankle HAL. The surface electrode sensors of the HAL were then attached to the TA and gastrocnemius muscles and the sensor cables were plugged into the control device. Afterward, the muscle activity of the TA and gastrocnemius muscles was observed on the monitor of the manual controller. The torque of the ankle HAL actuator was then adjusted and appropriately positioned in a dorsiflexed angle during ankle HAL training. Two therapists assisted the patient in wearing the ankle HAL, requiring approximately 3 min. The therapists monitored pain and device malfunction during ankle HAL training, and dorsiflexion training using the ankle HAL was implemented with periods of rest in between. Ankle dorsiflexion training using the ankle HAL was performed during hospitalization, two or three times a week. HAL training was performed in combination with conventional physical therapy five times per week.

Regarding physical evaluation, before and after the all-ankle HAL training session, the muscle strength of the right TA (measured using MMT), range of motion (ROM) of the right ankle dorsiflexion, and walking speed and step length in the 10 m walking test (10 MWT) were evaluated. The active and passive ROM of the right ankle dorsiflexion was manually measured using a goniometer. The 10 MWT was used to measure the gait speed and step length of the patient who used a walking device without the ankle HAL before and after all the training sessions. The walking speed (m/s) and time for the 10 MWT were measured using a handheld stopwatch. The muscle activities of the TA (healthy and affected sides) during ankle dorsiflexion without the ankle HAL were measured before and after training using a Trigo™ Lab wireless surface electromyography (EMG) system (Delsys Inc., Boston, MA, USA). This study was approved by the ethics committee of our university (TCRB18-38). The patient was informed about the aim and design of this study and provided written informed consent for participation and publication.

## 3. Results

Ankle dorsiflexion training using the ankle HAL was implemented two or three times per week for a total of 10 sessions (Figure 2a). The training period was 33–59 days (26 days) postoperatively. The time required for ankle HAL training (from wearing to the end of training) was approximately 30 min per session, without any adverse events, such as ankle-HAL-related pain or abrasions. The mean maximum dorsiflexion angle while wearing the ankle HAL was 5.0° ± 3.9° and the mean number of dorsiflexions per session was 218.0 ± 31.9. After 10 sessions of ankle HAL training, the right TA and EHL remained unchanged at MMT 1, and the right ankle dorsiflexion active and passive ROM were −50° to −50° and −15° to 0°, respectively. In the 10 MWT administered before and after ankle HAL training, gait speed and step length improved from 0.28 ± 0.04 m/s to 0.47 ± 0.04 m/s and 0.23 ± 0.02 m to 0.31 ± 0.02 m, respectively (Table 1). Figure 3a,b shows the muscle activities of the TA in left (healthy) and right (affected) ankle dorsiflexion, respectively, without the ankle HAL, before and after ankle HAL training. The muscle activities of the TA on the right (affected) (Figure 3b) were lower than those on the left (healthy) (Figure 3a) before and after ankle HAL training. In addition, the EMG wave of muscle activities of the TA on the right (arrows) was slightly clearer after than before ankle HAL training in the resting position immediately after ankle dorsiflexion (Figure 3a,b).

## 4. Discussion

This paper describes conservative treatment in the form of training using a novel robotics ankle HAL in a patient with foot drop due to peroneal nerve palsy after lumbar surgery. We developed a new robotics ankle HAL that can be used in training the ankle to safely perform dorsiflexion without any adverse effect even in a patient with ankle dorsiflexion muscle power MMT grade 1.

Effective conservative treatment for foot drop caused by peroneal nerve palsy that can promote improvement in ankle function has not been established. Some reports have suggested that functional electrical stimulation (FES) conservative treatment [14] is effective in managing foot drop [15,16]. However, FES treatment has not yet been established as an effective treatment method because it is not frequently used in clinical settings. Although orthotic treatment [17,18] for foot drop is effective in preventing ankle joint deformity and as an assistive walking device, it does not improve palsy. Therefore, effective conservative treatment for foot drop, which is caused by different factors such as peroneal neuropathy at the neck of the fibula, L5 radiculopathy, and stroke, has not yet been established. Although some patients with foot drop experience significant improvement, some permanently suffer from foot drop. Therefore, we developed a new robotics ankle HAL and investigated the safety and feasibility of ankle HAL training. In addition, this study is based on several research results for patients with hemiplegia caused by stroke. Therefore, we think that ankle palsy can improve by providing gait training using a type of HAL for both limbs [9,19,20,21].

The ankle HAL can be used for voluntary ankle joint training using surface bioelectrical signals (muscle action potentials) from the TA and gastrocnemius muscles. Improvement in passive dorsiflexion ROM was observed after ankle HAL training; however, no improvement in the active dorsiflexion ROM or in ankle dorsiflexion muscle power (TA and EHL) was observed. We think that voluntary active ankle dorsiflexion training using the ankle HAL is crucial in terms of errorless learning [22] and motor learning [23,24] in the field of neurorehabilitation. It repeatedly provided the patient with the correct motion for paralytic ankle dorsiflexion despite the low and weak muscle activities of the TA (Figure 3b). In addition, we think that the change from an unclear TA EMG wave (Figure 3b, left) to a clearer TA EMG wave (Figure 3b, right) was caused by a reduction in involuntary movements and an increase in voluntary movements. Although the patient could not properly contract her TA because of paralysis, we hypothesize that her ability to voluntarily move her TA was accelerated with ankle HAL training. Thus, we conjecture that ankle HAL training enhanced the patient’s motor learning. Although recovery from peripheral neuropathy is gradual, the interaction between the body and the environment continues during the recovery period. For this reason, recovering patients often learn incorrect movements. Treatment for peripheral neuropathy requires an induction of motor learning as well as restoration of muscle strength [23,24]. It is our opinion that training with the ankle HAL induced motor learning in the patient for the effective treatment of peroneal nerve palsy, a type of peripheral neuropathy.

This study should be repeated to evaluate the specific effect of ankle HAL training by including more patients with foot drop.

## 5. Conclusions

A new robotics ankle HAL was developed for patients with ankle-foot disabilities (foot drop) caused by common peroneal nerve palsy (after lumbar surgery) and stroke. Voluntary ankle dorsiflexion training using the novel robotics ankle HAL was safe and had no adverse effect when used in a patient with foot drop due to peroneal nerve palsy.

## Figures and Tables

**Figure 1 medicina-58-00395-f001:**
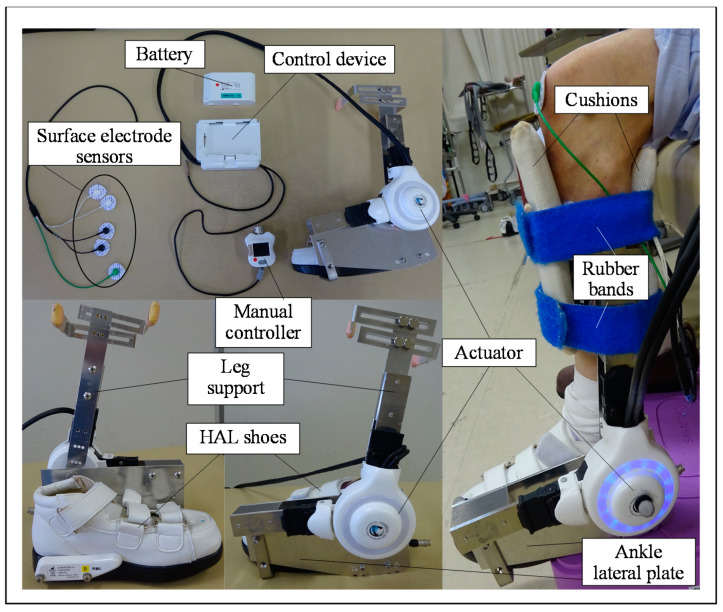
Structure of the left ankle HAL. The ankle HAL system consists of a control device, HAL shoe, leg support, ankle lateral plate, actuator, surface electrode sensor, manual controller, and battery.

**Figure 2 medicina-58-00395-f002:**
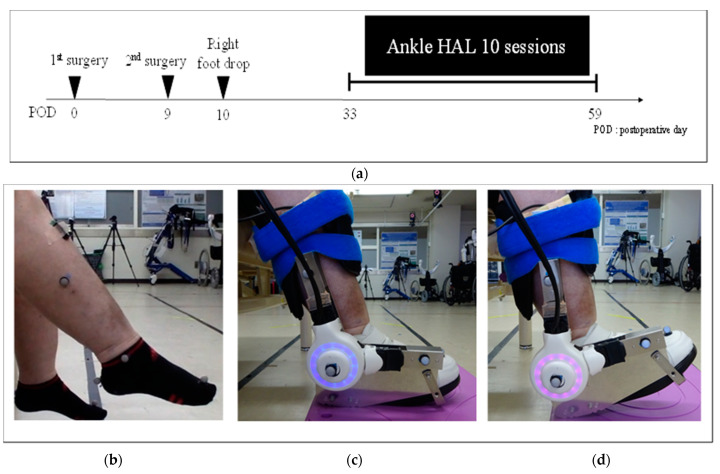
(**a**) Postoperative progress and duration of ankle HAL training; (**b**) right postoperative common peroneal nerve palsy with foot drop, maximum ankle dorsiflexion without ankle HAL on postoperative day 33 (ankle HAL session 1); (**c**) resting position of the ankle HAL (session 6); (**d**) maximum ankle dorsiflexion with ankle HAL (session 6).

**Figure 3 medicina-58-00395-f003:**
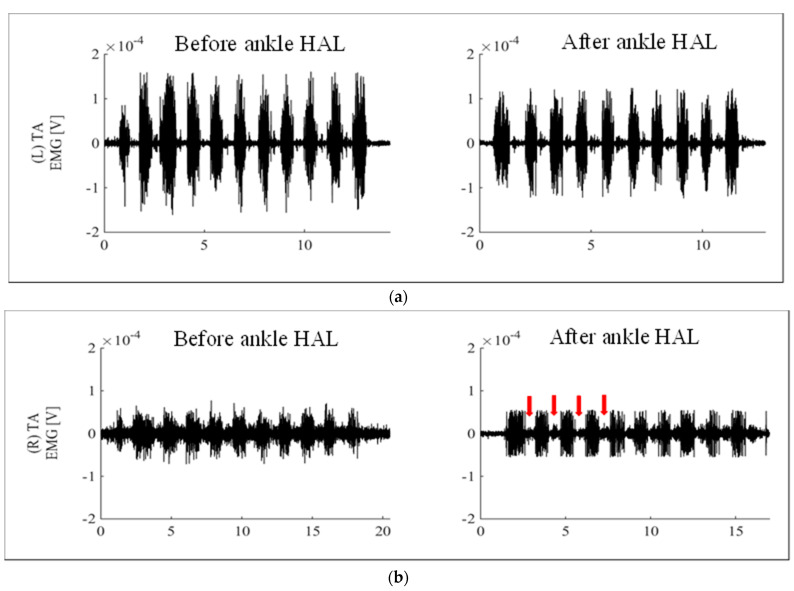
(**a**) Muscle activities of the TA on the left (healthy) ankle dorsiflexion before and after ankle HAL training. (**b**) Muscle activities of the TA on the right (affected) ankle dorsiflexion before and after ankle HAL training. Muscle activities of the TA on the right (affected) (**b**) were lower than those on the left (healthy) (**a**) before and after ankle HAL training. The EMG wave of muscle activities of the TA on the right after ankle HAL training was slightly clearer than that before ankle HAL training in the resting position immediately after ankle dorsiflexion (arrows).

**Table 1 medicina-58-00395-t001:** Results of dorsiflexion power (TA and EHL), dorsiflexion active and passive ROM, gait speed, and step length before and after ankle HAL training.

Parameter	Before Ankle HAL	After Ankle HAL
Dorsiflexion power (TA) (MMT)	1	1
Dorsiflexion power (EHL) (MMT)	1	1
Dorsiflexion active ROM (°)	−50	−50
Dorsiflexion passive ROM (°)	−15	0
Gait speed (m/s)	0.28 ± 0.04	0.47 ± 0.04
Step length (m)	0.23 ± 0.02	0.31 ± 0.02

## Data Availability

The original contributions presented in the study are included in the article/Appendix A; further inquiries can be directed to the corresponding author.

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
