# Peer review of "Development of a New Ankle Joint Hybrid Assistive Limb"

_medicina, 2022, doi:10.3390/medicina58030395_

Round 1

Reviewer 1 Report

This is an interesting and meaningful study describing a new ankle joint hybrid assistive limb. The manuscript was well-organized. However, some issues should be carefully addressed.

1. Line 66: “we added our original produced parts to …”

What are the original produced parts?

2. Line 185:

What does “active and passive ROM” mean?

3. Line 187-188: “gait speed and step length improved from 0.28 ± 0.04 m/s to 0.47 ± 0.04 m/s, 0.23 ± 0.02 m to 0.31 ± 0.02 m”.

Are there any significant differences in gait speed and step length between before and after ankle HAL training?

4. Line 188-194:

Are there any temporal or frequency-domain parameters of muscles activities? Only EMG of TA is shown in Figure 3, and does EMG of gastrocnemius muscle affected by the ankle HAL training?

5. Line 192-194: “Besides, the EMG wave of muscle activities of TA (after) in the right (arrow) was slightly clearer than that before ankle HAL training in resting of ankle dorsiflexion immediately”.

What does “clearer” mean?

6. Does the ankle HAL actively drive ankle’s dorsiflexion and plantar? Could authors show more details on the controlling system of the ankle HAL for example feedback system?

7. In my opinion, the effects of the ankle HAL on ankle motion and EMG had better be evaluated and discussed more deeply in the Section Discussion. Why is the ankle HAL considered to be effective? These information can provide theoretical supports for the future study.

8. Line 267-269: “Improvement in passive ROM of dorsiflexion was observed after ankle HAL training; however, no improvement in the active ROM of dorsiflexion and ankle dorsiflexion muscle power (TA and EHL) was observed”.

How to detect passive and active ROM of dorsiflexion?

9. Line 282: “Voluntary ankle dorsiflexion training using the novel robotics ankle HAL was safe”.

Why is “Voluntary ankle dorsiflexion training using the novel robotics ankle HAL” safe?

Author Response

Point-by-point response to the reviewer’s comments

Reviewer 1

Comments and Suggestions for Authors

This is an interesting and meaningful study describing a new ankle joint hybrid assistive limb. The manuscript was well-organized. However, some issues should be carefully addressed.

  1. Line 66: “we added our original produced parts to …”

What are the original produced parts?

Response:

The originally produced parts are the leg support and ankle lateral plate. We revised the manuscript, as below.

“...we added our originally produced parts (leg support and ankle lateral plate) to...” (Line 59)

  1. Line 185:

What does “active and passive ROM” mean?

Response:

Active ROM (range of motion) means the range of ankle dorsiflexion joint motion with the patient’s own power. On the other hand, passive ROM means the range of ankle dorsiflexion joint motion when moved by an external force (for example, evaluators).

  1. Line 187-188: “gait speed and step length improved from 0.28 ± 0.04 m/s to 0.47 ± 0.04 m/s, 0.23 ± 0.02 m to 0.31 ± 0.02 m”.

Are there any significant differences in gait speed and step length between before and after ankle HAL training?

Response:

We measured these values only twice at each time point using a handheld stopwatch (Lines 176-178). Therefore, although we could calculate the average and standard deviation gait speed and step length, we could not perform statistical analysis.

  1. Line 188-194:

Are there any temporal or frequency-domain parameters of muscles activities? Only EMG of TA is shown in Figure 3, and does EMG of gastrocnemius muscle affected by the ankle HAL training?

Response:

We performed EMG only for the TA, as it was the affected muscle in this case. The gastrocnemius muscle is responsible for plantar flexion during ankle HAL. When the patient tried to perform dorsiflexion on her own (she has palsy), the gastrocnemius muscle activity is detected with EMG, as ankle HAL would try to assist with plantar flexion. However, as we can adjust the torque with which the HAL provides assistance, we could set the HAL to not provide assistance with plantar flexion. Therefore, we believe that the EMG of the gastrocnemius muscle would not be uniquely affected during ankle HAL training.

  1. Line 192-194: “Besides, the EMG wave of muscle activities of TA (after) in the right (arrow) was slightly clearer than that before ankle HAL training in resting of ankle dorsiflexion immediately”.

What does “clearer” mean?

Response:

Thank you for your important remark. Although we already mentioned this aspect in the Discussion section, we realize that the explanation was insufficient. Therefore, we expanded the explanation, as below.

“The ankle HAL can be used for voluntary ankle joint training using surface bioelectrical signals (muscle action potentials) from the TA and gastrocnemius muscle. Improvement in passive ROM of dorsiflexion was observed after ankle HAL training; however, no improvement in the active ROM of dorsiflexion or in ankle dorsiflexion muscle power (TA and EHL) was observed. We believe that voluntary active ankle dorsiflexion training using the ankle HAL is crucial in terms of error-less learning [22] and motor learning [23, 24] in the field of neurorehabilitation. It repeatedly provided the patient with the correct motion for paralytic ankle dorsiflexion despite the low and weak muscle activities of the TA (Fig. 3b). In addition, we believe that the change from an un-clear TA EMG wave (Fig. 3b, left) to a clearer TA EMG wave (Fig. 3b, right) was caused by a reduction in involuntary movements and an increase in voluntary movements. Although the patient could properly contract her TA because of paralysis, we hypothesize that her ability to voluntarily move her TA was accelerated with ankle HAL training. Thus, we conjecture that ankle HAL training enhanced the patient’s motor learning. Although recovery from peripheral neuropathy takes time, the interaction between the body and the environment continues during the recovery period. For this reason, recovering patients of-ten learn incorrect movements. Treatment for peripheral neuropathy requires an induction of motor learning as well as restoration of muscle strength [23, 24]. It is our opinion that training with the ankle HAL induced motor learning in the patient for the effective treatment of peroneal nerve palsy, a type of peripheral neuropathy.” (Lines 250-269)

  1. Does the ankle HAL actively drive ankle’s dorsiflexion and plantar? Could authors show more details on the controlling system of the ankle HAL for example feedback system?

Response:

The ankle HAL can be used to assist a patient with ankle dorsiflexion and plantar flexion motions in real time by using the wearer’s muscle action potential. The actuator consists of an angular sensor that is placed on the lateral ankle joint. The bioelectrical signals (muscle action potentials) are detected through the electrodes on the anterior and posterior surface of the wearer’s lower leg. These various biologic signals and information, detected by the angular sensor, are processed by a control device. The HAL has a feedback system in that patients can view their own bioelectrical signals on a monitor in real time during ankle HAL training. Therefore, we added the sentence below to the text.

“The actuator consists of an angular sensor on the lateral ankle joint. These various biologic signals and information detected by the angular sensor are processed by a control device (computer). The HAL provides the wearer with a feedback system, allowing them to view their own bioelectrical signals on a monitor in real time during ankle HAL training.” (Lines 81-85)

  1. In my opinion, the effects of the ankle HAL on ankle motion and EMG had better be evaluated and discussed more deeply in the Section Discussion. Why is the ankle HAL considered to be effective? These information can provide theoretical supports for the future study.

Response:

Thank you for your important suggestion. Please see our revision of the text as detailed in our response to comment 5 above.

  1. Line 267-269: “Improvement in passive ROM of dorsiflexion was observed after ankle HAL training; however, no improvement in the active ROM of dorsiflexion and ankle dorsiflexion muscle power (TA and EHL) was observed”.

How to detect passive and active ROM of dorsiflexion?

Response:

We manually measured the passive and active ROM of ankle dorsiflexion by using a goniometer. As this was not explained in our original manuscript, we added the following sentence to the Materials and Methods section of the revised manuscript.

“The active and passive ROM of the right ankle dorsiflexion was measured manually using a goniometer.” (Lines 157-158)

  1. Line 282: “Voluntary ankle dorsiflexion training using the novel robotics ankle HAL was safe”.

Why is “Voluntary ankle dorsiflexion training using the novel robotics ankle HAL” safe?

Response:

We monitored the patient for any adverse events during each ankle HAL session. As a result, we demonstrated that ankle HAL training is feasible without any adverse events, such as ankle HAL-related pain or abrasions (Lines 170-172). Therefore, we concluded that voluntary ankle dorsiflexion training using the novel robotics ankle HAL was safe and feasible in this case report.

Reviewer 2 Report

The Paper presents a novel ankle hybrid assistive limb for patients with foot drop due to common peroneal nerve palsy or stroke. A case study is presented and the results are discussed in terms of improvement (or missing improvement) on several indexes. 

The strength of the presented work: development of the working device, proved ready for use and the exposition of  how to use such device in rehabilitation. Voluntary movement can be triggered by muscle activation  notwithstanding the nervous issue underlying the problem. In general the paper describes well the experiments and the results and, in my opinion, qualifies for publication.

Limitations: the human test is just a case study so general conclusions are not yet possible. The presented scenario, i.e. training the ankle joint, does not tell much about the potential or the limitation of integrating the ankle joint in walking and/or with the rest of the HAL robot. (from the paper I understand that gait was tested just without exoskeleton as a test before and after the treatment, if it is not the case specify it explicitly in the results).

Minor remarks:
- i would not use the word "cyborg" for an exoskeleton, "robot" would be more appropiate
- fig 1 "battery" is misspelled
- line 103 "weaner" instead of "wearer" 

Author Response

Point-by-point response to the reviewer’s comments

Reviewer 2

Comments and Suggestions for Authors

The Paper presents a novel ankle hybrid assistive limb for patients with foot drop due to common peroneal nerve palsy or stroke. A case study is presented and the results are discussed in terms of improvement (or missing improvement) on several indexes.

The strength of the presented work: development of the working device, proved ready for use and the exposition of how to use such device in rehabilitation. Voluntary movement can be triggered by muscle activation notwithstanding the nervous issue underlying the problem. In general the paper describes well the experiments and the results and, in my opinion, qualifies for publication.

Limitations: the human test is just a case study so general conclusions are not yet possible. The presented scenario, i.e. training the ankle joint, does not tell much about the potential or the limitation of integrating the ankle joint in walking and/or with the rest of the HAL robot. (from the paper I understand that gait was tested just without exoskeleton as a test before and after the treatment, if it is not the case specify it explicitly in the results).

Response:

Thank you for the accurate summary of our manuscript.

As you stated, our conclusion in this paper is not yet a general conclusion, as it is a single case report and not a study to determine the effectiveness of ankle HAL training on patients with common peroneal nerve palsy and stroke. Therefore, we corrected the conclusion, as follows.

“Voluntary ankle dorsiflexion training using the novel robotics ankle HAL was safe and had no adverse effect when used in a patient with foot drop due to peroneal nerve palsy.” (Lines 275-276)

We do not think that the ankle joint should be integrated with the rest of the HAL robot during walking, as each HAL robot type has its own aim, such as for both limbs, lateral limb, a single joint (knee or elbow), and the lumbar type.

As you mentioned above, gait was only tested without the exoskeleton ankle HAL, before and after ankle HAL training.

Minor remarks:

- i would not use the word "cyborg" for an exoskeleton, "robot" would be more appropiate

- fig 1 "battery" is misspelled

- line 103 "weaner" instead of "wearer"

Response:

- We have corrected the word “cyborg” to “robot” throughout the manuscript.

- We have corrected the typographical error in the figure.

- We have corrected the word “weaner” to “wearer” throughout the manuscript.

Reviewer 3 Report

(1)The abstract is a little bit long.
(2)This paper has somewhat application values, but lacks of academic values. What are the difficulties and how to solve them?
(3)Section 1 is too short. At the end of Section 1, “there has been no report on the development of ankle joint HAL.” What are the relationships between the existing methods and this paper? What are the new problems and difficulties of ankle joint HAL? This paper only reports a simple development if there are no problems or difficulties.
(4)The authors only stated what they did in this paper. They should clearly how they designed the algorithm, and why / when the steps can play a role.
(5)The length of this paper is short. More experiments, comparisons, discussions can be added.

Author Response

Point-by-point response to the reviewer’s comments

Reviewer 3

Comments and Suggestions for Authors

  • The abstract is a little bit long.

Response:

We deleted some of the information on the ankle HAL system from the abstract.

  • This paper has somewhat application values, but lacks of academic values. What are the difficulties and how to solve them?

Response:

We think that our paper has academic value in that our case report is an introduction to the newly developed rehabilitation robot and a demonstration of its feasibility. If the ankle HAL robot proves effective for the treatment of ankle disabilities (foot drop or hemiplegia due to stroke) in future investigations, it will be of great value to patients and physicians alike.

  • Section 1 is too short. At the end of Section 1, “there has been no report on the development of ankle joint HAL.” What are the relationships between the existing methods and this paper? What are the new problems and difficulties of ankle joint HAL? This paper only reports a simple development if there are no problems or difficulties.

Response:

Thank you for your important suggestions. This is the first report of the use of the ankle HAL to treat a patient. The problem with ankle joint and foot disabilities, particularly foot drop, is that it is permanent in some patients because of common peroneal nerve palsy. The same is true for stroke; most patients with stroke have permanently ankle-foot disabilities such as ankle and foot spasticity. Therefore, there is an urgent need to ameliorate palsy of the ankle joint for foot drop due to peroneal nerve palsy, to increase the dorsiflexion muscle strength of the ankle joint, and to reduce ankle joint spasticity for ankle joint disorders caused by stroke. We expect that the ankle HAL will solve these problems.

  • The authors only stated what they did in this paper. They should clearly how they designed the algorithm, and why / when the steps can play a role.

Response:

This case report is the first stage of the introduction of the newly developed ankle HAL. Next, we plan to increase the number of patients (case series), and thereafter, to proceed with controlled studies (with conventional rehabilitation).

  • The length of this paper is short. More experiments, comparisons, discussions can be added.

Response:

In our original manuscript, we discussed conservative treatment such as FES (functional electrical stimulation) treatment and orthotic treatment. In the revised manuscript, we added more information to the discussion, as below.

“Although recovery from peripheral neuropathy takes time, the interaction between the body and the environment continues during the recovery period. For this reason, recovering patients often learn incorrect movements. Treatment for peripheral neuropathy re-quires an induction of motor learning as well as restoration of muscle strength [23, 24]. It is our opinion that training with the ankle HAL induced motor learning in the patient for the effective treatment of peroneal nerve palsy, a type of peripheral neuropathy.” (Line 263-269)

Round 2

Reviewer 1 Report

Thank you for your thoughtful reply. The manuscript has been improved a lot after the revision. I believe the paper is almost suitable for publication.

One last thing, are there any temporal or frequency-domain parameters of TA muscles activities? I think the quantitative parameters (for example: integrated electromyography) are better than only EMG activity figures.

Author Response

* Author's Notes to Reviewer

Author's Reply to the Review Report (Reviewer 1)

Review Report Form

Comments and Suggestions for Authors

Thank you for your thoughtful reply. The manuscript has been improved a lot after the revision. I believe the paper is almost suitable for publication.

One last thing, are there any temporal or frequency-domain parameters of TA muscles activities? I think the quantitative parameters (for example: integrated electromyography) are better than only EMG activity figures.

*Reply to Reviewer 1

Response

Thank you for your important suggestion. We think that the quantitative parameters of EMG are the mean amplitude and integrated EMG (iEMG), which you mentioned in you comment. We believe think that the mean amplitude is a better quantitative parameter than the iEMG because the iEMG is time dependent. However, in this study, we wanted to demonstrate the TA muscle activities in the resting position; therefore, we chose the low EMG wave in Figure 3 (a) and (b) in this case report. As you mentioned, we believe that both the mean amplitude and the iEMG are better than only the EMG activity values for readability.

Reviewer 3 Report

The length is still too short for a journal paper.
This paper has somewhat application values, but still lacks of academic values.
This paper lacks of sufficient analysis and quantitative comparisons.

Author Response

* Author’s Notes to Reviewer

Author’s Reply to the Review Report (Reviewer 3)

Review Report Form

Comments and Suggestions for Authors

The length is still too short for a journal paper.

This paper has somewhat application values, but still lacks of academic values.

This paper lacks of sufficient analysis and quantitative comparisons.

Response

Thank you for your comments. The academic value of this case report is the world’s first case report of ankle HAL. Many patients suffer from ankle disorders with associated foot drop due to common peroneal nerve palsy and the sequelae of stroke. In the future, ankle HAL will be introduced not only in Japan but also in several facilities around the world. As you mentioned, it may have little academic value, but we do not believe that there is no academic value in this case report.